# Effect of water activity on rates of serpentinization of olivine

Hector M. Lamadrid[1,2], J. Donald Rimstidt[1], Esther M. Schwarzenbach[3], Frieder Klein[4], Sarah Ulrich[1], Andrei Dolocan[5] & Robert J. Bodnar[1]

The hydrothermal alteration of mantle rocks (referred to as serpentinization) occurs in submarine environments extending from mid-ocean ridges to subduction zones. Serpentinization affects the physical and chemical properties of oceanic lithosphere, represents one of the major mechanisms driving mass exchange between the mantle and the Earth's surface, and is central to current origin of life hypotheses as well as the search for microbial life on the icy moons of Jupiter and Saturn. In spite of increasing interest in the serpentinization process by researchers in diverse fields, the rates of serpentinization and the controlling factors are poorly understood. Here we use a novel *in situ* experimental method involving olivine micro-reactors and show that the rate of serpentinization is strongly controlled by the salinity (water activity) of the reacting fluid and demonstrate that the rate of serpentinization of olivine slows down as salinity increases and $H_2O$ activity decreases.

[1] Department of Geosciences, Virginia Tech, Blacksburg, Virginia 24061, USA. [2] Department of Earth Sciences, University of Toronto, 22 Russell Street, Toronto, Ontario, Canada M5S 3B1. [3] Institute of Geological Sciences, Freie Universität Berlin, 12249 Berlin, Germany. [4] Woods Hole Oceanographic Institution, Marine Chemistry and Geochemistry Department, Woods Hole, Massachusetts 02543, USA. [5] Texas Materials Institute, The University of Texas at Austin, Austin, Texas 78712, USA. Correspondence and requests for materials should be addressed to H.M.L. (email: hm.lamadrid@utoronto.ca).

Serpentinization encompasses a series of hydration reactions that occur when ultramafic rocks are exposed to circulating aqueous fluids at temperatures lower than ~400 °C, leading to the formation of serpentine phases (lizardite and chrysotile) ± brucite ± talc ± magnetite, among other minerals[1]. Serpentinization affects the chemical composition, rheology, magnetic properties, seismic structure and habitability of the shallow lithosphere at slow- and ultraslow-spreading mid-ocean ridges, continental margins and forearc settings of subduction zones[2–8]. Serpentinization also influences subduction related processes[9] and the geochemical cycling of volatile species (that is, $H_2O$, $CO_2$ and $H_2S$) and fluid-mobile elements[10–13]. Recent findings suggest that serpentinization of olivine-rich lithologies also takes place on other planetary bodies, such as the icy moons of Jupiter and Saturn, which in turn, has important implications concerning their habitability[14,15].

Despite the pivotal role that serpentinization plays in a number of geological and biological processes and its central role in current origin of life hypotheses[16–19], few experimental studies have attempted to determine the rates of serpentinization reactions and the rates that have been reported diverge widely[20–25]. Furthermore, the environmental factors that affect the reaction rates are incompletely constrained. In the present work, we used synthetic fluid inclusions (SFIs)[26,27] as micro-reactors in olivine crystals to monitor the serpentinization reaction with time. We trapped fluid with different initial salinities and followed reaction progress at serpentinization conditions (280 °C). The results show that the rates of olivine serpentinization are strongly influenced by the aqueous fluid salinity. The micro-reactor technique presents several advantages and permits monitoring mineral precipitation and water activity *in situ* and in real time.

## Results

**SFI preparation**. We trapped SFIs[26,27] in gem quality (inclusion-free) natural olivine at 650 °C and 5.5 kbar, conditions where olivine and $H_2O$ coexist stably (Fig. 1). The starting $H_2O$-NaCl-$MgCl_2$ fluids in each experiment had a Na/Mg molar ratio similar to that of seawater (8/1) and total salinities of 1, 3.5, 6 and 10 wt%. The capsules were sealed with an arc welder and then placed into cold-seal pressure vessels and held at the desired trapping conditions (650 °C and 5.5 kbar) for 30 days, which allowed the olivine host to anneal and trap some of the fluid as SFIs. The *P–T* conditions of trapping of the SFI were chosen such that the trapped fluid would have a density ~0.8 g cm$^{-3}$. Following inclusion synthesis, doubly polished wafers were prepared and individual wafers were placed into a furnace at 280 °C and ambient external pressure (~1 bar)—at this temperature, olivine is no longer stable and will react with the aqueous solution to produce serpentinization. At a temperature of 280 °C and constant volume of the SFI, the calculated pressure inside the SFI is ~500 bar.

The SFIs can be envisioned as micro-reactors, a reaction vessel in which material can only be exchanged between the aqueous solution and the surrounding host olivine. A major advantage of this approach over most other existing experimental techniques is that changes in mineralogy and fluid composition can be monitored simultaneously in real time as the reaction progresses.

**Serpentinization experiments**. Following the formation of the SFI in olivine, samples were heated to 280 °C and held at this temperature for 30 to 270 days. For each fluid composition and salinity, several SFI in each olivine wafer were selected to monitor serpentinization reaction progress (Table 1). Each inclusion was photographed and mapped, and each was regularly monitored

optically, by microthermometry and by Raman spectroscopy. The polished olivine wafers were briefly (2–5 h) removed from the furnace every 5 days, examined petrographically and analysed by Raman spectroscopy, and then returned to the furnace. Figure 1 (0 days) shows a typical SFI trapped in olivine immediately before starting the heating experiments to initiate the serpentinization reaction.

**Reaction progress**. After a few days of reaction time, small clusters of secondary minerals are observed in the SFI (Fig. 1, 15 days) and the size and number of individual crystals in the SFI progressively increased with time until the inclusions were nearly filled with solids, rendering them opaque (Fig. 1, 45 days).

The onset of the serpentinization reaction varied with the salinity of the starting fluid. Reaction products were first observed after 5 days in ~50% of the SFIs with a starting salinity of 1 wt%. The proportion of the SFI showing evidence of reaction after 5 days decreased to ~20, ~5 and 0% in the SFI with starting salinities of 3.5, 6 and 10 wt%, respectively. At low salinities the proportion of SFI showing reaction products increased rapidly with time until they were ubiquitous in the SFI. For the highest salinity experiments (10 wt%), the first evidence of reaction product formation was observed only after >120 days.

In addition to the experiments in which samples were regularly removed from the furnace for observation and analysis, two samples containing SFI with salinities of 1 and 3.5 wt% were maintained at constant temperature using a Chaixmeca microscope heating stage[28]. For these samples, the SFI could be monitored without having to cool the sample from 280 °C to room temperature. These experiments were conducted to confirm that repeatedly cycling the samples from 280 °C to room temperature and back to 280 °C did not significantly affect the progression of the experiments. No differences were observed between the thermal cycling experiments and the constant-temperature Chaixmeca experiments.

**Mineral characterization**. Raman spectroscopic analyses of the SFI (Fig. 2a,b) confirmed that the reaction products consist of serpentine minerals (Srp, lizardite and chrysotile) and brucite (Brc) produced according to the simplified reaction (note that Fe can substitute for Mg in all three mineral phases, but the reaction below is written in terms of the Mg endmembers):

$$2Mg_2SiO_4(Ol) + 3H_2O \Leftrightarrow Mg_3Si_2O_5(OH)_4(Srp) + Mg(OH)_2(Brc) \tag{1}$$

Magnetite is a minor but common byproduct of serpentinization reactions. It forms when the $Fe^{2+}$ released from olivine is oxidized in the fluid and precipitates magnetite and generates $H_2$:

$$3Fe^{2+} + 4H_2O \Leftrightarrow Fe_3O_4 + H_2 + 6H^+ \tag{2}$$

In our experiments, magnetite was present in detectable amounts only after >120 days of reaction time. Small magnetite crystals, formed via reaction (2) (but not detectable by optical microscopy) should begin to precipitate inside the micro-reactors soon after reaction (3) starts and the solubility of magnetite in the solution is exceeded. However, we interpret the lack of observable magnetite in the inclusions early in the experiments to reflect the fact that the crystals are not optically resolvable (~0.5 μm) by the methods used here (including scanning electron microscopy (SEM) and transmission electron microscopy (TEM) in Fig. 2c,d). In order for magnetite to become sufficiently large that they can be recognized either optically or during microanalysis, hydrogen must be lost from the SFI. The relatively long reaction time before magnetite is recognizable with optical microscopy in the SFI is attributed to its relatively sluggish precipitation, which may be

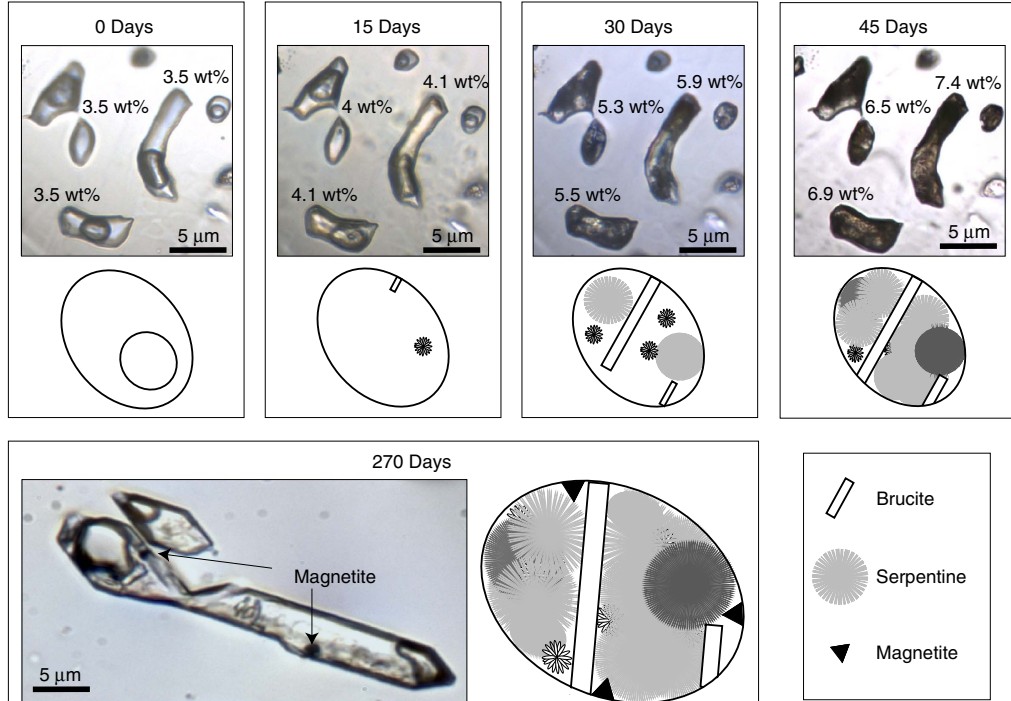

**Figure 1 | Photomicrographs and schematic illustrations of the evolution of the micro-reactors through time.** Photomicrographs of SFI taken before the serpentinization reactions were initiated (0 days) and after 15, 30, 45 and 120 days of reaction time at 280 °C. The images show the same group of SFIs that had an initial salinity of 3.5 wt%. The numbers next to individual SFIs in the 15-, 30- and 45-day images represent the measured salinity in wt%. The image for 120 days shows magnetite in an SFI that had an initial salinity of 1 wt%. The reaction progress is shown schematically below the photomicrographs. Before reaction the fluid inclusions only contain an aqueous solution and a vapor phase. After 15 days at 280 °C, small crystals of brucite and serpentine appear. After 30 days, more crystals have formed. After 45 days, the fluid inclusion is nearly filled with crystals. After 120 days, magnetite crystals appear.

**Table 1 | Overview of experimental conditions and alteration mineralogy.**

| Run | Fluid composition | S | Experiment type | T (°C) | t (days) | Alteration mineral assemblage |
|---|---|---|---|---|---|---|
| 1 | $H_2O$-NaCl-$MgCl_2$ | 1 | Furnace | 280 ($\pm$5) | 120 | S + B |
| 2 | $H_2O$-NaCl-$MgCl_2$ | 1 | Chaixmeca | 280 ($\pm$8) | 120 | S + B + M |
| 3 | $H_2O$-NaCl-$MgCl_2$ | 3.5 | Furnace | 280 ($\pm$5) | 30 | S + B |
| 4 | $H_2O$-NaCl-$MgCl_2$ | 3.5 | Furnace | 280 ($\pm$5) | 60 | S + B |
| 5 | $H_2O$-NaCl-$MgCl_2$ | 3.5 | Furnace | 280 ($\pm$5) | 90 | S + B |
| 6 | $H_2O$-NaCl-$MgCl_2$ | 3.5 | Chaixmeca | 280 ($\pm$8) | 120 | S + B + M |
| 7 | $H_2O$-NaCl-$MgCl_2$ | 6 | Furnace | 280 ($\pm$5) | 120 | S + B |
| 8 | $H_2O$-NaCl-$MgCl_2$ | 10 | Furnace | 280 ($\pm$5) | 270 | S + B + M |

S, salinity in wt%.
Furnace, experiments in which sample was heated in a furnace and removed from the furnace and quenched to room temperature at various times for Raman and microthermometric analysis. As such, the sample was cycled from room temperature to 280 °C numerous times during the experiment.
Chaixmeca, experiments in a microscope heating stage whereby the SFI were kept at a constant temperature of 280 °C and the heating stage was placed onto the Raman microprobe at various times for analysis.
Mineral assemblages: B, brucite; M, magnetite; S, serpentine.

further limited by competition for iron by serpentine and brucite, or by the slow rate at which hydrogen diffuses out of the inclusions[29] when (or if) the inclusion approaches equilibrium.

**Mineral composition**. The chemical composition of the reaction products was inferred from elemental maps obtained from time-of-flight secondary ion mass spectrometry (TOF-SIMS) analyses of the SFI (Fig. 3 and Supplementary Tables 1 and 2). The ratio of Mg to Fe, defined as the Mg number (Mg# = $\frac{Mg}{Mg+Fe}$ * 100), of brucite is 87–88, whereas serpentine phases have an Mg# ranging from 90 to 94. These values are similar to those of secondary minerals produced during serpentinization of olivine reported from other experimental studies[25].

## Discussion
The serpentinization reaction, even in its simplest form (equation (1)), is the result of several simultaneous reactions that can be summarized as follows. First, olivine dissolves:

$$Mg_2SiO_4(Ol) + 4H^+ = 2Mg^{2+} + SiO_{2(aq)} + 2H_2O \qquad (3)$$

When the concentrations of dissolved species become sufficiently high, serpentine and brucite nucleate and precipitate according to:

$$3Mg^{2+} + 2SiO_{2(aq)} + 5H_2O = Mg_3Si_2O_5(OH)_4(Srp) + 6H^+ \qquad (4)$$

$$Mg^{2+} + 2H_2O = Mg(OH)_2(Brc) + 2H^+ \qquad (5)$$

Reactions (4) and (5) consume water and generate hydrogen ions

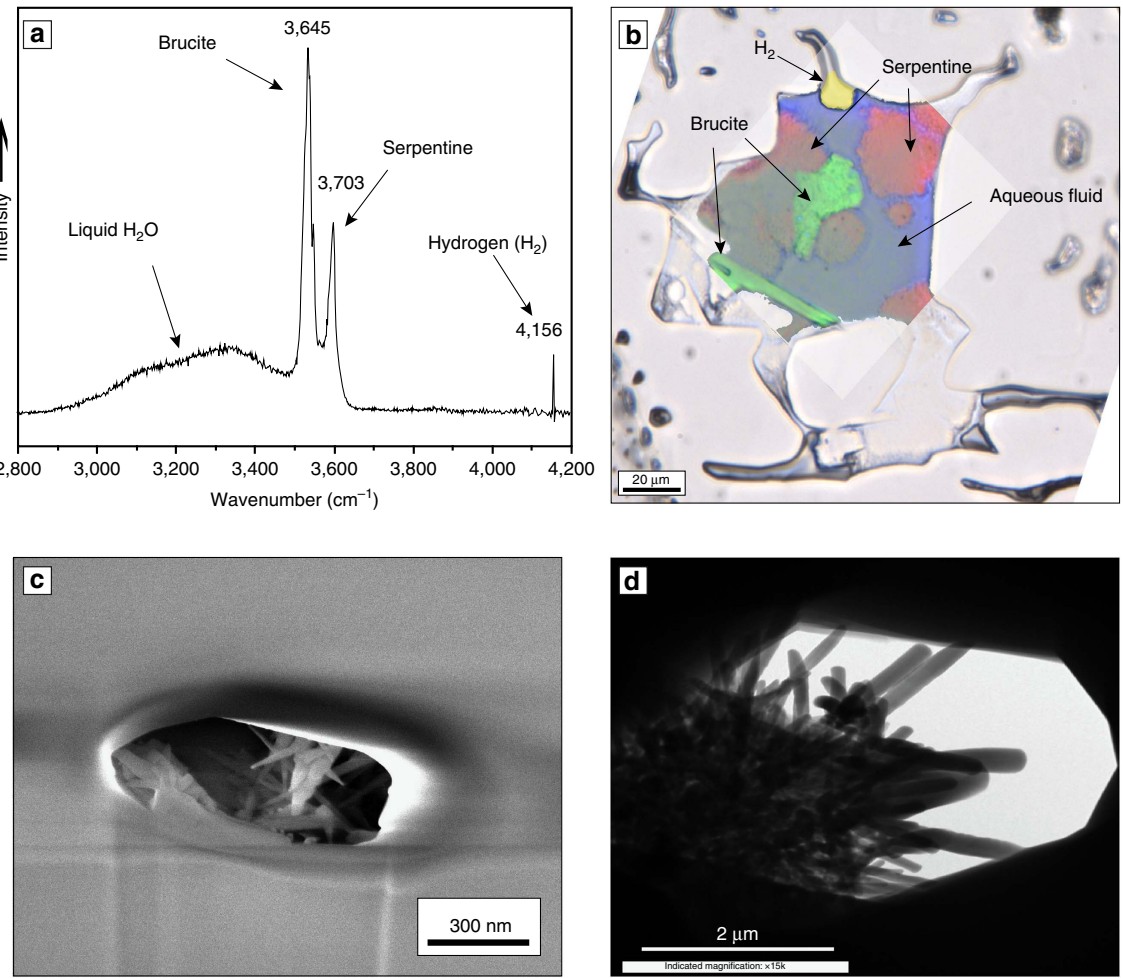

**Figure 2 | Characterization of the reaction products.** (**a**) Raman spectrum showing characteristic O–H bands of serpentine (lizardite) and brucite, the broad band for liquid $H_2O$ and $H_2$ in the vapour bubble. (**b**) Raman map of the SFI: serpentine shown in red, brucite in green, aqueous solution in blue and $H_2$ in yellow. (**c**) Reaction products attached to the wall of an SFI imaged after the SFI was excavated by FIB. (**d**) TEM image of an SFI showing the fibrous and conical shapes of serpentine (chrysotile) crystals.

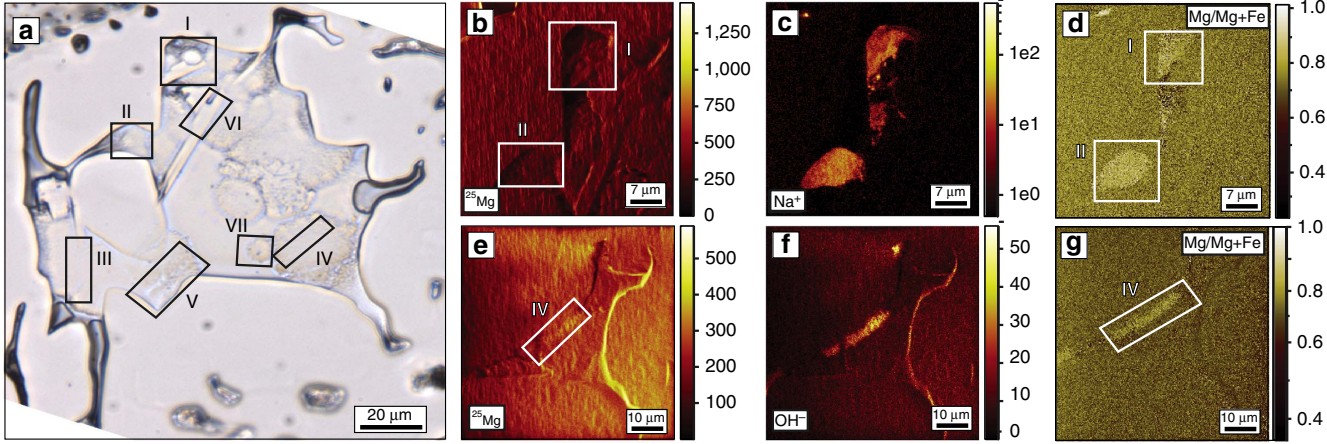

**Figure 3 | TOF–SIMS elemental maps of the SFI from Fig. 2b.** (**a**) Photomicrograph of the SFI. (**b–g**) Elemental maps of different regions of the SFI, labelled I to VII, to show the ROI (regions of interest) where the TOF–SIMS analyses were performed (Supplementary Table 2). (**b–d**) $^{25}Mg^+$, $Na^+$ and $^{25}Mg/(^{25}Mg + Fe)$ maps showing compositional differences between the host and the reaction products, labeled as ROI I and II. The $^{25}Mg/(^{25}Mg + Fe)$ map is obtained from the secondary ion yield map $^{25}Mg^+/(^{25}Mg^+ + Fe^+)$ by using a multiplication factor that ensures the olivine background yields a ratio of 0.8. (**e–g**) $^{25}Mg^+$, $OH^-$ and $^{25}Mg/(^{25}Mg + Fe)$ maps showing compositional differences between the host and the reaction products labeled as ROI IV. The $Na^+$ is contained in the frozen aqueous phase in the SFI. (**f**) $OH^-$ maps showing the presence of serpentine and $H_2O$ ice. (**g**) $^{25}Mg/(^{25}Mg + Fe)$ map of ROI IV.

($H^+$) that in turn promote forsterite dissolution (reaction 3). Eventually, the overall process reaches a steady state whereby the rates of Mg and $SiO_{2(aq)}$ released from olivine dissolution (reaction 3) are balanced by their rates of consumption by serpentine (reaction 4) and brucite (reaction 5) formation.

As the formation of serpentine and brucite consumes $H_2O$, the salinity of the aqueous solution in SFI increases as the reaction proceeds. We used the increase in salinity as a proxy for $H_2O$ consumption and reaction progress, to quantify the reaction rates (mol m$^{-2}$ s$^{-1}$) (Fig. 4a,b, Supplementary Table 3 and Supplementary Data 1 and 2). The reaction progress is expressed as the extent of reaction ($\xi$), defined as the number of moles ($n$) of reactant consumed or product generated per unit time and normalized by the stoichiometric coefficient ($v$) for the phase being consumed or produced. Therefore, the reaction rate of a species $i$ is defined as the time ($t$) derivative of the extent of reaction:

$$\frac{d\xi}{dt} = \frac{1}{v_i}\frac{dn_i}{dt} \tag{6}$$

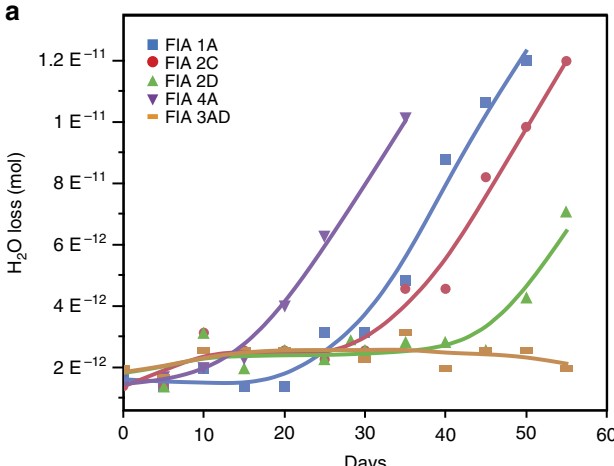

**a**

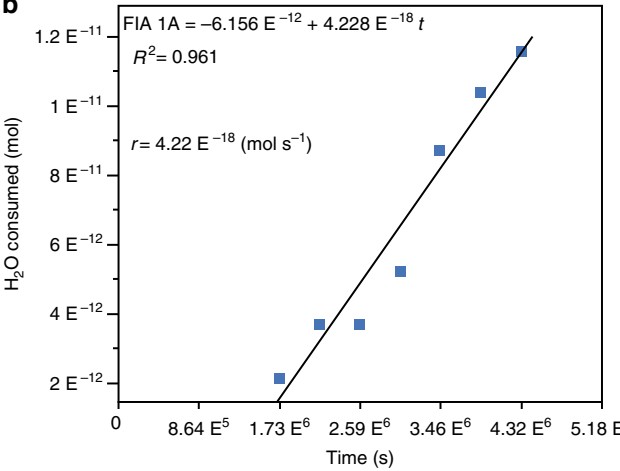

**b**

FIA 1A $= -6.156\,E^{-12} + 4.228\,E^{-18}\,t$

$R^2 = 0.961$

$r = 4.22\,E^{-18}$ (mol s$^{-1}$)

**Figure 4 | Reaction progress as a function of $H_2O$ consumed through time.** (**a**) Example showing the moles of $H_2O$ consumed as a function of time for five SFI micro-reactors from Run 4 (3.5 wt% $H_2O$-NaCl-MgCl$_2$ starting salinity). Solid lines are drawn as references to identify data from individual SFI. (**b**) Moles of $H_2O$ consumed versus time for SFI FIA1A from run 4. The initial rate method was applied for each SFI starting from the time at which the reaction started. The slope of the fitted line represents the rate of reaction ($r$).

As such, the reaction rate can be defined in terms of the rate of change in the amount of any of the species that participate in the reaction, that is,

$$\frac{d\xi}{dt} = -\frac{1}{2}\frac{dn_{W}}{dt} = -\frac{1}{3}\frac{dn_{W}}{dt} = \frac{1}{1}\frac{dn_{Srp}}{dt} = \frac{1}{1}\frac{dn_{Bru}}{dt} \tag{7}$$

where the subscripts Fo, W, Srp and Bru correspond to the phases forsterite (olivine), water, serpentine and brucite, respectively. Figure 5a shows the rate of serpentinization as a function of the activity of water. The average rates of reaction were determined to be $2.71 \times 10^{-8}$, $1.17 \times 10^{-8}$, $3.37 \times 10^{-9}$ and $2.86 \times 10^{-10}$ mol m$^{-2}$ s$^{-1}$ for the SFIs with starting salinities of 1, 3.5, 6 and 10 wt%, respectively (Supplementary Table 3). The results show that serpentinization rates decrease rapidly with increasing salinity (or decreasing activity of water), and agree with previous studies that have reported that dissolution rates of forsterite at laboratory conditions (25 °C and 1 bar) decrease as $a_{H_2O}$ decreases[30]. A rate equation was generated by fitting the log of the reaction rates as a function of log $a_{H_2O}$ with a second-order polynomial:

$$\log J = -7.24(\pm 0.03) + 103(\pm 4)\log a_{H_2O} - 3481(\pm 4)$$
$$(\log a_{H_2O} - 0.0081)^2 \tag{8}$$

where $-7.24$ is the log of the rate constant at 280 °C and the coefficient 103 represents the apparent reaction order of $H_2O$. The serpentinization rates obtained here agree with the rates of reaction from Wegner and Ernst[21] after extrapolating the results to unit activity (pure water) with a log $J$ of $-7.51$ (Fig. 5b).

These results document the dramatic effect of salinity, as a proxy for water activity $a_{H_2O}$, on the rate of serpentinization of olivine and are consistent with previous studies that have suggested that salinity can have a significant effect on dissolution kinetics[31–33] including the dissolution of forsterite at far from equilibrium conditions[30,34]. The effect of water activity $a_{H_2O}$ on the reaction rate is consistent with the generalized serpentinization reaction given by equation (1), in which the driving force for production of serpentine and brucite is inversely proportional to the cube of the activity of $H_2O$ and is also consistent with the dissolution of olivine (equation (3)) being the rate controlling step of the reaction[23].

These results provide a geochemical basis to support the previously recognized notion that pervasive serpentinization of oceanic lithosphere (which is dominated by the mineral olivine) requires open-system behaviour, as imprinted by the seawater-dominated isotope signature commonly observed in serpentinized peridotites[35–38]. In particular, this requires continual influx of a lower salinity aqueous fluid (seawater) to dilute the serpentinization fluid and allow serpentinization of olivine to continue[39]. Moreover, salinity may be a major rate-limiting factor where fluid influx is restricted due to the absence of open fractures and fluid migration proceeds along grain boundaries or within nanoscale porosity[40–42]. Furthermore, the effect of salinity on the serpentinization process places constraints on the environments in which serpentinization is likely to occur on Earth and possibly on other planetary bodies where high salinity fluids are believed to exist[43–45].

The micro-reactor technique used in this study provides a novel and promising tool to monitor fluid-rock reactions *in situ* and in real time and can be applied to a wide variety of host minerals, reaction products, temperatures and different starting fluid compositions (for example, CaCl$_2$, CO$_2$, SO$_4$ and Al$_2$O$_3$).

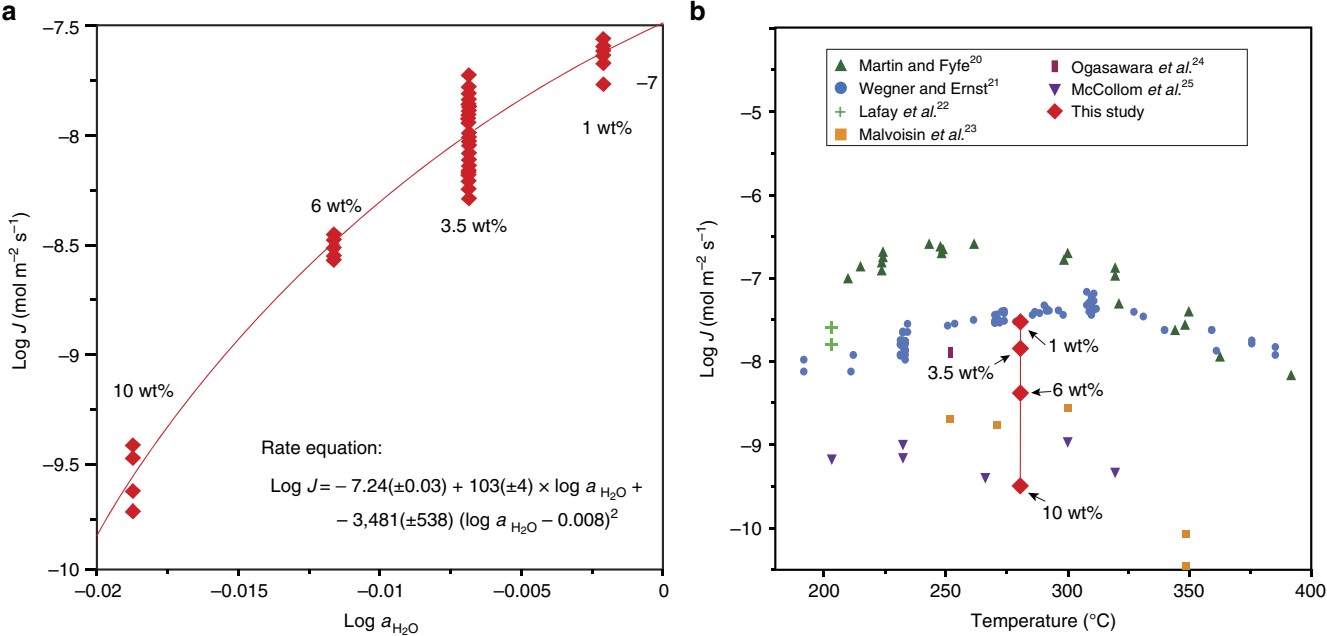

**Figure 5 | Experimentally determined serpentinization rates.** (**a**) Logarithm of the serpentinization rate (red diamonds) as a function of the logarithm of the activity of water. Each red diamond represents the result obtained from an individual SFI. The rate equation predicts a log rate constant of − 7.24 and an apparent reaction order of 103. (**b**) Comparison of results of this study at 280 °C with experimentally determined reaction rates from the literature[20–25] plotted as a function of temperature. Most of the previous experimental studies have used pure water[20,21,23,24] and a few others used different fluid compositions[22,23,25].

## Methods

**SFI preparation.** SFIs were trapped in gem quality (inclusion free) natural olivine crystals (Fo85) at 650 °C and 5.5 kbar following procedures described by Sterner and Bodnar[26], and similar to Bali et al.[27]. The crystals were thermally fractured by heating to 700 °C in an oven at atmospheric pressure and then quenching by dropping the crystals into cold distilled water. The crystals were dried overnight in a vacuum oven at 110 °C. Next, one or two 3–5 mm in diameter crystals of olivine were loaded into a platinum capsule (2.5 × 0.5 mm) together with ∼100 µl of an $H_2O$-NaCl-MgCl$_2$ fluid (Na/Mg weight ratio of 8/1) with salinities of either 1, 3.5, 6 or 10 wt%. The capsules were sealed with an arc welder. Platinum capsules were placed into cold-seal pressure vessels and held at 650 °C and 5.5 kbar for 30 days, which allowed the olivine host to anneal and trap some of the fluid it was in contact with as synthetic secondary fluid inclusions that were then used as micro-reactors in this study. After the fluid inclusion synthesis, the capsules were cooled to room temperature and the crystals removed from the capsules and mounted in thermal cement for cutting (∼100 to 200 µm wafers) and polishing. The sizes of the SFI varied between ∼1 and 40 µm, with most being ∼10 µm in diameter.

**Raman characterization.** Raman analyses were performed using a JY Horiba LabRam HR (800 mm) spectrometer, 600 grooves per mm grating, slit width of 150 µm and the confocal aperture was set at 100 µm. A 514.529 nm (green) Laser Physics 100S-514 Ar + laser was used with a 50 mW output power at the source and ∼10 mW at the sample, focused through either a ×40 or ×100 objective, and an electronically cooled open electrode 1,024 × 256 pixel CCD (charge-coupled device) detector. Raman X–Y mapping was conducted using a JY Horiba LabRam HR confocal Raman spectrometer with a 473 nm laser source and a 600 grooves per mm grating and a 100 µm confocal aperture and ×100 objective.

**Electron microprobe analyses.** The composition of the olivine host crystal (Supplementary Table 1) used in the experiments was determined using a Cameca SX-50 electron microprobe (EMPA). Accelerating potential was 15 kV, 20 nA current and 1 µm beam size, using natural and synthetic mineral standards for calibration.

**Mineral and chemical characterization.** To expose the reaction products in the SFI to obtain SEM images, we used a Helios 600 NanoLab FIB-SEM. Chemical analyses of the contents of the SFI after the serpentinization experiments were conducted using TOF-SIMS with a TOF.SIMS 5 IONTOF instrument[46–48]. At each step during the analysis, the ion beam consisting of Bi$_1^+$ ion pulses (30 keV ion energy) was first set in the high current bunched mode (20 ns pulse duration, ∼3.7 pA measured sample current, ∼1 µm lateral resolution) to determine the masses of interest with high mass resolution. The beam was typically raster-scanned over areas of 100 × 100 µm or 200 × 200 µm depending on the region of interest. Then, the beam was set in the burst alignment mode (100 ns pulse duration, ∼0.03 pA measured sample current, ∼200 nm lateral resolution) to produce high spatial resolution maps of the selected regions for all masses of interest. To expose the SFI in the olivine hosts at various depths we used an $O_2^+$ sputtering ion beam (1 keV ion energy, ∼300 nA measured sample current) that was raster-scanned over an area of 500 × 500 µm and centred over the region of interest. The resulting sputtering rate was calculated at 0.3 nm s$^{-1}$ by using a Wyko NT 9100 optical profilometer to estimate the sputtered crater depth. The sample was maintained at a base temperature of − 100 °C at all times during the analysis by inductive liquid nitrogen cooling to prevent the water from desorbing from the exposed SFI. A constant current (21 eV electron energy) electron beam was shot on the sample during data acquisition to reduce sample charging. The mass resolution was > 3,000 and > 100 (m dm$^{-1}$) for the high current bunched and burst alignment modes, respectively. The TOF-SIMS provided ratios of the number of counts for one mass (isotope) to the number of counts for some other isotope and these counts were converted into a concentration ratio. To do this, the count ratio for Mg$^+$ and Fe$^+$ were converted into concentration ratios using the known Mg to Fe concentration ratio of the olivine host obtained from EMPA and the count ratio obtained by TOF-SIMS analyses for this same area of the crystal (see Supplementary Table 2).

**TEM analyses.** The sample was polished to a thickness of 30 µm using an Allied Multiprep with 3M Imperial diamond lapping papers of 30, 15, 6, 3 and 1 µm grit sizes. After polishing, the sample was mounted on a copper aperture grid with epoxy. Then the sample was milled in a Fischione 1,010 low-angle argon ion mill at ∼10 µm h$^{-1}$ for several hours until the SFI were exposed. Areas immediately adjacent to the hole became thin enough to be electron transparent. The sample was placed in a JEOL single tilt holder. TEM images were collected at Virginia Tech on a JEOL 2100 TEM operated at 200 kV, using a Gatan Orius CCD camera.

**Monitoring reaction progress.** The salinity of the SFIs was monitored by measuring the freezing point depression (FPD) of the aqueous phase (temperature of melting of the last ice crystal, $T_m$) using a Linkam THMSG 600 °C heating and freezing stage. During the serpentinization reaction $H_2O$ was removed from the solution and incorporated into the reaction products, whereas Na$^+$ and Cl$^-$ remained in the solution. Trace amounts of Cl$^-$ can be incorporated into serpentine, substituting for OH$^-$ (0.1 to 0.4 wt%), but this is considered to have a negligible effect on salinity in the experiments. Each measurement was corrected for analytical error by measuring the $T_m$ of a pure $H_2O$ SFI. The correction factor ranged from 0.8 to 1.0 °C and the precision of the FPD measurements is ∼ ± 0.1 °C. The salinity (wt% eq. NaCl) as a function of the FPD was calculated using FPD salinity equation[49]. The triple point of pure $H_2O$ occurs at 0.01 °C and

0.006 bars, where liquid $H_2O$, vapour $H_2O$ and solid $H_2O$ (in this case, the Ice I polymorph) occur in equilibrium. With increasing salt concentration, the triple point migrates to lower temperature and this effect is referred to as the FPD of the solution. FPD is a colligative property that depends only on the ratio of the number of moles of the solute ($NaCl$–$MgCl_2$ in our experiments) to that of the solvent $H_2O$. Thus, as $H_2O$ was transferred from the aqueous solution into product phases, the ratio of moles of solute to moles of solvent ($H_2O$) increased and the temperature of the triple point decreased. The relationship between the salinity (moles of solute in the solution) and FPD has been determined using data for the ternary system $H_2O$-$NaCl$-$MgCl_2$ (refs 49,50).

In our experimental method, we quantify the rates of serpentinization based on the amount of water removed from the aqueous solution and incorporated into hydrous phases. We use the FPD of the aqueous solution as a proxy for the amount of water removed from solution and the precision of our FPD measurement is $\pm 0.1\,^{\circ}C$. However, some amount of $H_2O$ must be removed from the solution and incorporated into hydrous phases before the salinity is increased by an amount sufficient to produce a $0.1\,^{\circ}C$ lowering of the FPD. The actual amount of $H_2O$ that must be removed from solution to produce a $0.1\,^{\circ}C$ lowering of the FPD is also a function of the starting salinity. Thus, for example, if the initial salinity is 1 wt% NaCl (FPD = $-0.6\,^{\circ}C$), $\sim 15\%$ of the $H_2O$ in the initial fluid must be consumed by hydration reactions to drive the FPD lower by $0.1\,^{\circ}C$ to $-0.7\,^{\circ}C$ (corresponding to a salinity of $\sim 1.2$ wt% NaCl). However, the same $0.1\,^{\circ}C$ lowering of the FPD for a starting salinity of 3.5 wt% NaCl requires the consumption of 4.7% of the $H_2O$ in the initial fluid, 6 wt% NaCl initial salinity requires a consumption of 2.6% of $H_2O$ and 10 wt% NaCl initial salinity requires consumption of only 1.4% of the initial $H_2O$, as the FPD is lowered from $-6.6\,^{\circ}C$ (10 wt% NaCl) to $-6.7\,^{\circ}C$ (10.1 wt% NaCl). Thus, until a sufficient amount of reaction has occurred to consume enough $H_2O$ to lower the FPD by $\geq 0.1\,^{\circ}C$, in the absence of observed reaction products in the inclusions we have no way of knowing a priori that the reaction has started and, depending on the starting salinity, the amount of $H_2O$ consumed (and, therefore, the amount of reaction that must occur) could require from $\sim 1$ to 15% of the initial $H_2O$ to be removed from solution. All of the microthermometric data (FPD, salinity and $H_2O$ moles consumed) are provided in Supplementary Data 1.

**Salinity changes and $H_2O$ consumed during serpentinization.** To calculate the mass of $H_2O$ consumed by the serpentinization reaction as a function of the FPD, we constructed a simple mass balance model that relates the mass of $H_2O$ (moles) removed from the aqueous fluid during the reaction to the FPD. The mass of $H_2O$ (g) contained in a 10 µm diameter SFI was calculated for $0.1\,^{\circ}C$ increments of FPD as follows. The total mass of solution in the SFI is given by:

$$M_{SFI} = (V_{SFI})(\rho_{SFI}) \qquad (9)$$

where $V_{SFI}$ is the volume of an ideal spherical fluid inclusion ($5.24 \cdot 10^{-10}\,cm^3$) and $\rho_{SFI}$ is the density of the solution in the SFI (in $g\,cm^{-3}$). The density of the aqueous solution in the SFI was calculated using the HOKIEFLINCS H2O_NaCl programme[51] and assuming that the system $H_2O$-$NaCl$ provides a reasonable approximation for the relationship between salinity and density for the fluids in the SFI. Accordingly, the densities at $650\,^{\circ}C$ and 5.5 kbar for salinities of 1, 3.5, 6 and 10 wt% NaCl are $\sim 0.84$, $\sim 0.87$, $\sim 0.88$ and $\sim 0.90\,g\,cm^{-3}$, respectively. The masses of NaCl and $H_2O$ in the SFI are given by:

$$M_{NaCl} = (X_{NaCl})(M_{SFI}) \qquad (10)$$

$$M_{H_2O} = (1 - X_{NaCl})(M_{SFI}) \qquad (11)$$

where $M_{NaCl}$ and $M_{H_2O}$ are the masses of NaCl and $H_2O$, respectively, in the SFI in grams and $X_{NaCl}$ is the mass fraction of NaCl (wt% NaCl/100) in the solution.

$$M_{H_2O} = \frac{M_{NaCl} - (X_{NaCl}M_{NaCl})}{X_{NaCl}} \qquad (12)$$

where $M_{H_2O}$ is the mass of $H_2O$ (g), $M_{NaCl}$ is the mass of NaCl (g) and $X_{NaCl}$ is salinity of the fluid in terms of weight fraction of NaCl.

The amount of $H_2O$ consumed by the reaction is estimated from the measured FPD as follows. First, the mass of $H_2O$ in the SFI corresponding to the initial salinity and measured FPD is calculated according to equation (12). Then, the amount of $H_2O$ in the SFI after some amount of reaction has occurred to lower the FPD by some measureable amount (corresponding to an increase in salinity) is calculated. The loss of $H_2O$ from the solution to the product phases corresponding to a given change in salinity was then calculated:

$$M_{w\,loss} = M_{w,0} - M_w \qquad (13)$$

where, $M_{w,0}$ is the mass of $H_2O$ in grams at $t = 0$ and $M_w$ is the mass of $H_2O$ obtained from equation (9). The number of moles of $H_2O$ consumed $n_w$ was then calculated as:

$$n_w = M_{w\,loss}(g) \times \frac{1\,mol}{18\,g} \qquad (14)$$

For every initial salinity (1, 3.5, 6 and 10 wt%), the number of moles of $H_2O$ consumed was fit as a polynomial expression to determine the number of moles of $H_2O$ consumed as a function of the salinity of the SFI. For every FPD measurement in an individual SFI, the number of moles of $H_2O$ leaving the system was calculated. All calculations and procedures are provided in Supplementary Data 2.

**Rates of reaction and rate equation.** Reaction progress was followed in several dozen SFI by monitoring the change in the FPD through time. To follow the reaction progress it was necessary to observe the changes inside the SFI. As reaction progressed, the amount of reaction products increased rendering the inclusion opaque and challenging our ability to follow the reaction progress at high extents of reaction. Moreover, as the $H_2O$ inside the SFI is consumed by reaction (1), the internal pressure and volume, as well as the concentrations and compositions inside the inclusion, will be modified from the initial conditions. To avoid the problem of visibility and the effects of changing of the fluid compositions and pressure inside the SFI micro-reactors, our observations where analysed using the initial rate method[52]. Batch experiments usually require data collection that includes large extents of reaction to circumvent issues related to modifications of the initial conditions (that is, composition, pH, pressure and so on)[53]. Alternatively, the initial rate method uses numerous short-term experiments with a small number of concentration measurements at equally spaced times. Then, the concentration data versus time data for each experiment are fitted to a function and the slope of that function (rate) is extrapolated to time = 0. In our experiments, at time zero little or no brucite + serpentine had formed and no significant water was consumed. The small volume and pressure changes associated with this small extent of reaction did not affect the result. As such, in our experiments, the number of moles of $H_2O$ consumed by the serpentinization reaction with time was found from the change in $n_w$ with time:

$$r = \frac{\Delta n_w}{\Delta t} \qquad (15)$$

where $r$ is the rate of $H_2O$ consumption in $mol\,s^{-1}$, $\Delta n_w$ is the number of moles of $H_2O$ consumed by the transformation reaction and $\Delta t$ is the time in seconds. The number of moles of $H_2O$ removed from solution and incorporated into hydrous phases as a function of time ($t$) for every SFI was fit by a linear regression and the rate of reaction was taken as the slope of the regression line[53,54] (see Fig. 4).

As the reaction occurs at an interface that separates two phases (solid and liquid), the rate expresses how fast a component is transferred to or from that interface and the area of the interface must be taken into account as follows:

$$J = \frac{r}{A} \qquad (16)$$

where $J$ is the flux of $H_2O$ from the liquid phase into the product phases, defined as the rate of $H_2O$ consumption per unit surface area ($mol\,m^{-2}\,s^{-1}$), $r$ is the rate of $H_2O$ consumption in $mol\,s^{-1}$ and $A$ is the surface area of a 10 µm spherical fluid inclusion ($1.26 \times 10^{-9}\,m^{-2}$) (Supplementary Data 2). To generate rate equations, rates obtained as described above were converted into rate equations that summarize the effects of solution compositions[52]. These experimental rates were fit to an equation that relates the rate to the concentration ($a$, activity) of $H_2O$, $i$, raised to a power ($n$).

$$r = k \prod a_i^{n_i} \qquad (17)$$

This equation was transformed to a second order polynomial form:

$$\log r = \log k + \sum n_i \log a_i + \sum n_i (\log a_i)^2 \qquad (18)$$

The data were fit to an equation of this form using a polynomial function to determine the rate constant ($k$) and the apparent reaction order ($n$). The activities ($a_i$) used in this study were calculated with the software code EQ3/6 (ref. 55) using the thermodynamic database for 50 MPa as described in Klein et al.[56]. Data for the rates, fluxes, activities and extent of reaction of all the experiments are provided in Supplementary Table 3.

**Serpentinization rates from the literature.** To compare the serpentinization rates obtained in this study with the rate data available in the literature, we converted the published data[20–25,57] into fluxes ($mol\,m^{-2}\,s^{-1}$) by applying the Shrinking Particle Model[52,58]. This model assumes that all particles are spheres and the rate of dissolution equals the rate of conversion. As such, the conversion rates (olivine to serpentine and brucite, per day from the literature) were converted into fluxes ($mol\,m^{-2}\,s^{-1}$) by determining the rate constant following expressions:

$$1 - (1-\alpha)^{1/3} = k_p t \qquad (19)$$

where $\alpha$ is the conversion rate fraction, $k_p$ is the rate constant and $t$ is time (s).

$$k_+ = \frac{R_0 k_p}{V_m} \qquad (20)$$

where $k_+$ is the flux ($mol\,m^{-2}\,s^{-1}$), $R_0$ is the particle size radius at time zero and $V_m$ is the molar volume of olivine. $R_0$ was obtained by using the averaging method of Tester et al.[52,59], which finds the effective diameter of the grain sizes by the following method:

$$D_e = \frac{D_{max} - D_{min}}{\ln\left(\frac{D_{max}}{D_{min}}\right)} \qquad (21)$$

where $D_e$ is the effective diameter (m), $D_{max}$ is the maximum grain size and $D_{min}$ is the minimum grain size diameter.

**Data availability.** The data that support the findings of this study are available from the corresponding author upon reasonable request.

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

## Acknowledgements

We thank Charles Farley for help with Raman analyses; Chris Winkler, Rui Serra Maia and Jay Tuggle at the Nanoscale Characterization and Fabrication Laboratory (NCFL) for help with TEM and FIB SEM. This material is based upon work supported by the National Science Foundation under Grant OCE-1459433 to R.J.B. and E.M.S. Consejo Nacional de Ciencia y Tecnología (CONACyT), the Virginia Tech Department of Geosciences and Virginia Tech Graduate School provided partial funding to HML during this study. F.K. was supported by The Andrew W. Mellon Foundation Endowed Fund for Innovative Research. We also acknowledge the NSF grant DMR-0923096 used to purchase the TOF-SIMS instrument at Texas Materials Institute, UT Austin.

## Author contributions

H.M.L. developed the concept of SFIs as micro-reactors and performed the experiments reported in this communication. J.D.R. assisted in the interpretation of the kinetic data and geological application of the data. E.M.S. assisted in the interpretation and geological application of the data. F.K. carried out the Raman mapping as well as assisted in the interpretation and geological application of the data. S.U. performed sample preparation and TEM analyses and interpretation of the data. A.D. performed sample preparation for TOF-SIM analyses and interpretation of the data. R.J.B assisted in the experimental design, interpretation and geological application of the data. All authors discussed the results, commented and participated on the writing of the manuscript.

## Additional information

**Competing interests:** The authors declare no competing financial interests.

**Publisher's note**: 

