## [Peer Review File · Nature Communications]

Reviewers' Comments:

Reviewer #1 (Remarks to the Author)

This study provides new experimental insights into serpentinization processes. The experimental set is innovative and deal with reactions occurring in synthetic fluid inclusions (SFI), olivine reactant being the fluid host. The study provides interesting results and new ideas on the effect/understanding of a key parameters - water activity - on the serpentinization reaction kinetic.

- The manuscript is well written, the choice of this experimental set is well defined (allows to stop and restart experiments; simultaneous in-situ fluid and mineralogy measurements). The methods section is for me complete.
- The study overall is well thought and provide first evidence of salinity effect on reaction kinetic (even if this can be calculated). Here salt mixing in water is used as a proxy to test the effect of a change of H₂O activity on the rate of serpentinization inside SFI.
- The range of salinity tested is relevant for earth science system. The temperature tested is relevant for ultramafic rocks hydrothermal alteration in oceanic context.
- Based on their observations, the authors propose a rate equation as a function of water activity which, for me, is supported by sufficient experimental results.
- The study might interest every person working on the understanding of serpentinization reaction (experimental part, forces, parameters, scale... and natural part) and people who take into account experimental results in studying natural case (both in hydrothermal and metamorphic systems + in extra-terrestrial systems).
- The previous studies are discussed but sometimes not ideally chosen (e.g. line 38 about serpentinization influence on subduction related processes)
- The discussion about magnetite in the experiments is not clear. It seems that magnetite are only observed after 120 days of reaction time. The Mg# of reaction product is strongly lower than the one of olivine and Fe in solution should be very low in experienced conditions. Herein magnetite is likely to be present. Authors mention that H₂ increasing preclude magnetite precipitation... this explain magnetite precipitation to stop but not the absence of magnetite before.

In the Methods section

- The methods are well described and I estimate that it has sufficient detail, so that experiments and characterization can be reproduced.
- The discussion about uncertainties of the method to determine H₂O activity is not sufficiently detailed.
e.g. what is the repercussion of an error of 0.1°C for FPD measurements (looking at supplementary table, comparing FIA1G and FIA1H SFI a difference of 0.1°C in TM change H₂O loss by a factor 1)
- From what I understand, all SFI are assumed to be 10µm in diameter (impossible to measure single SFI volume), this is important to mention it more clearly as SFI initial volume might affect the rate at which salinity increases in fluid - even if it doesn't change measured salinity and inferred H₂O activity -. For a single sample even if initial salinity is the same in all SFI and reaction operate in the same way, olivine dissolution (related to surface, i.e. to V of SFI) remains a limiting factor (even when the steady state between dissolution and precipitation is reached), in turn H₂O consumption and salinity at a given time will be different due to different SFI initial volume and different "bulk SFI" reaction rate.
- Supplementary data are for me complete, but a detail of calculations for "H₂O consumed" in supplementary excel sheet will be appreciated.
- Figures and table are well chosen and well represent observations. Figure 1 schematic representation (1B) can have a better design with legend on one side and enlarged scheme. The legend can refer to Figures 2 and

3 as minerals fig. 1A are hard to see.

The summary of previous experimental studies is appropriate and well illustrate the importance of fluid salinity (as a proxy of water activity) on serpentinization kinetic at 280°C.

The conclusion could mention the need to tests other temperatures, other water activities or the effect of other diluents (e.g. CO₂, SO₄...).

I recommend to publish this manuscript.

Dr. Romain Lafay

Reviewer #2 (Remarks to the Author)

General comments

The study deals with the use of a novel and promising method to monitor fluid-mineral interactions in-situ. The method is based on the use of aqueous-fluid inclusions as micro-reactors. The authors have had the great idea to use the freezing point depression (FPD) of the fluid inclusion to derive its water content from salinity. Basically, if the host mineral is able to react with the water from the inclusion, as it is the case here for olivine forming hydrous phases (i.e., serpentine and brucite), reaction progress/kinetics can be inferred from time-resolved FPD measurements along with some extend of modeling. In comparison with conventional batch experiments performed with sieved mineral grains, the present method avoids, in particular, artifacts associated with the presence of tiny little grains sticking onto the surface of the larger ones and which, consequently, escape sieving. These unexpected small grains often control reaction rates at low reaction extents.

This ingenious methodology has been applied to olivine serpentinization kinetics under increasing salinity (NaCl) from 1 to 10 wt.% at 280°C and ca. 500 bars. The authors observed a clear decrease in the serpentinization kinetics towards higher salinity / lower water activity. They therefore entitled their paper: "Effect of water activity on rates of serpentinization of olivine";. They conclude that serpentinization of oceanic lithosphere requires open-system behavior, i.e. with a reacting fluid which is renewed, otherwise serpentinization will ceased due to increasing salinity and decreasing serpentinization rates.

1/ whereas I am very enthusiastic about the experimental method and its novelty, I find this main conclusion of the paper somehow trivial. Indeed, there are several pieces of evidence that serpentinization of the oceanic lithosphere is an open-system chemical process which is often described as an "alteration" process. Sr isotopic ratio (⁸⁷Sr/⁸⁶Sr), $\delta^{11}\text{B}$ or Li isotopes as well as changes in major element composition in serpentinites (e.g., Boschi et al. 2008, GCA 72, 1801-1823; Vils et al., EPSL 386, 414-425; Decitre et al., 2002, G3 3; Malvoisin, 2015, EPSL 430, 75-85) all support significant chemical transfer between "circulating" seawater and serpentinizing abyssal peridotites. The challenge today seems rather to quantify the magnitude of chemical transfers, e.g., through the derivation of water to rock ratio (W/R). Actually, one could argue that even in the hypothetical case where water would not be renewed in the serpentinization process, irrespective of the effect salinity on reaction kinetics, serpentinization would cease after the consumption of all the water. Therefore I would suggest that the authors use their remarkable dataset to put constraints on minimum W/R values for serpentinization to proceed at a realistic timescale (e.g., Früh-Green et al., 2003, Science 301, 495-498).

2/ the authors nicely decomposed olivine serpentinization process into successive dissolution steps, the first of which being olivine dissolution (Reaction 3). However, recently, Olsen et al. (2015, JGR Planets 120, 388-400) showed how increasing ionic strength, IS (or lowering water activity) slows down olivine dissolution kinetics. The findings by Olsen et al. (2015) allow therefore to anticipate the effect of increasing IS on serpentinization kinetics as described here since olivine dissolution is expected to be the serpentinization rate-limiting step (e.g., Malvoisin et al., 2012).

The results obtained here must be discussed in light of those published data.

More specific comments

1/ Reaction progress and pressure change

The authors based their kinetic approach on reaction rates $(\Delta n_w)/\Delta t$ what is obviously find. However, it would have been useful to present the kinetics data in terms of reaction progress as well. Basically, how much of the initial water content of the inclusion has been consumed over the reaction duration? Based on the initial inclusion volume of $5.24 \cdot 10^{-10}$ cc (Line 362) with a density of 22.4 cc/mol (280°C/500 bar, supcrt92), there is about $2.3 \cdot 10^{-11}$ mole H₂O per inclusion (NaCl content is not considered here). A water consumption of $1.2 \cdot 10^{-11}$ mole as shown on extended data Fig.1A corresponds therefore to about 50% reaction progress, meaning that about half of the initial water has been consumed.

Another parameter of primary interest is the volume change of the inclusion as a function of reaction progress. Considering the molar volume @ 500 bar/280°C of the phases involved in Reaction (1), i.e., forsterite (43.8 cc/mol), brucite (24.6 cc/mol), chrysotile (108.5 cc/mole) and water (22.4 cc/mol), the volume change of reaction (1) is $-(21.7/3)$ cc per mole of H₂O consumed whereas the solid volume change is $+(45.5/3)$ cc/mole consumed H₂O (see Hanley, 1992, Geology 20, 705-708). For 50% reaction, the inclusion volume will decrease by $(45.5/3)1.2 \cdot 10^{-11} = 1.82 \cdot 10^{-10}$ cc, i.e. 35%. Accordingly, if 50% of the starting water has been consumed and if the inclusion volume has only decreased by 35%, this means that internal pressure has also decreased. Recalling that H₂O is liquid and thus very little compressible, we can anticipate a significant pressure drop, possibly down to the liq-vap curve! This very rough calculation is certainly not accurate but still the relationship between reaction progress, volume change and internal pressure must at least be discussed. The non-isobaric conditions in the micro-reactor may appear as a limitation of the technique.

2/ Effect of salinity

Another issue that could have been addressed with the present database is whether olivine dissolution kinetics differs in seawater and pure water. This is important since olivine serpentinization kinetics has been investigated experimentally in the past using either pure water or synthetic seawater. Here, the authors used a linear log-log regression to their H₂O flux - water activity data (Fig. 4A). Extrapolation of the regression to $a_{H_2O} = 1$ leads to a $\log J$ of -7.76 which, at 280°C, would be consistent with Wegner and Ernst's dataset rather than with that of Malvoisin et al. (2012) as claimed Line 127 p. 6. A way to reconcile the most recent kinetics dataset with the present one would be to consider a non-linear regression, with $\log J = -8.5/-9.0$ at $a_{H_2O} = 1$. Then, differences between serpentinization kinetics in seawater (3.5 wt.%) and in pure water would cancel.

Reviewer #3 (Remarks to the Author)

Review of manuscript NCOMMS-16-22733-T by Lamadrid et al.

This manuscript provides rates of olivine serpentinization as a function of fluid salinity, a parameter that has not been investigated before, most of the previous studies using either pure water or seawater salinities. The experimental protocol used here is original, as it associates previously developed techniques (synthetic fluid inclusion formation, salinity measurements classically used for fluid inclusion studies, optical/Raman monitoring used e.g. in diamond anvil cells, and final FIB-TEM/SEM and Raman characterization of reaction products). TOF-SIMS is also used and provides valuable chemical analyses that locate the different minerals and the fluid within the fluid inclusion, as it may have been done with Raman mapping.

The paper is concise and well written. Results are undoubtedly of good quality, consistent and

provide new data on serpentinization rate that complement previously published ones. But despite the strong interest of the community for serpentinization, the conclusions on the role of water activity on serpentinization remains limited as water is an essential ingredient to this hydration reaction and its activity is of course determinant. Implications on the required open-system behavior and the fresh fluid renewal at olivine interface, notably by cracking/fracturation, are not very new neither. This has been proposed by many other authors from various types of studies (Jamveit et al., Andreani et al. 2007, Plumper et al., 2012; Godard et al, 2013; Klein et al. 2015, ...) - it is already very well established. The technical development, while elegant, is inspired from previous methods and does not appear sufficient to justify the publication of this paper in a journal as Nature Comm.

I also have some other questions/comments on the manuscript that deserve improvements:

- How is the volume of fluid inclusion measured? Not clear.
- The reactive surface area is calculated for a spherical inclusion which is obviously not the case according to images. What is the error on rate calculations?
- Line 66: Why only 50% of the SFI reacted after 15 days for a salinity of 1 wt% if they are all similar? What happened in the others? Reaction is delayed?
- Raman identification of hydrogen indicate a gaseous phase, right? Is it only due to cooling during measurement? Or do you also see it in your isothermal experiments?
- Line 83-87: Why is it necessary to remove hydrogen? Is the equilibrium reached? Does this sentence result from any calculation, e.g. based on magnetite grain size over hydrogen solubility and FI volume, or other?
- How is hydrogen retained in some natural fluid inclusions if it already diffuses at lab timescale?
- It would be interesting on figure 4 to report on the salinity of the previous experiments for a direct comparison.

Reviewers' comments:

Reviewer #1 (Remarks to the Author):

This study provides new experimental insights into serpentinization processes. The experimental set is innovative and deal with reactions occurring in synthetic fluid inclusions (SFI), olivine reactant being the fluid host. The study provides interesting results and new ideas on the effect/understanding of a key parameters - water activity - on the serpentinization reaction kinetic.

- The manuscript is well written, the choice of this experimental set is well defined (allows to stop and restart experiments; simultaneous in-situ fluid and mineralogy measurements). The methods section is for me complete.
- The study overall is well thought and provide first evidence of salinity effect on reaction kinetic (even if this can be calculated). Here salt mixing in water is used as a proxy to test the effect of a change of H₂O activity on the rate of serpentinization inside SFI.
- The range of salinity tested is relevant for earth science system. The temperature tested is relevant for ultramafic rocks hydrothermal alteration in oceanic context.
- Based on their observations, the authors propose a rate equation as a function of water activity which, for me, is supported by sufficient experimental results.
- The study might interest every person working on the understanding of serpentinization reaction (experimental part, forces, parameters, scale... and natural part)and people who take into account experimental results in studying natural case (both in hydrothermal and metamorphic systems + in extra-terrestrial systems).
- The previous studies are discussed but sometimes not ideally chosen (e.g. line 36 about serpentinization influence on subduction related processes)

Reference was replaced by a more recent and appropriate reference in Line 37 (Hirth and Guillot, 2013).

- The discussion about magnetite in the experiments is not clear. It seems that magnetite are only observed after 120 days of reaction time. The Mg# of reaction product is strongly lower than the one of olivine and Fe in solution should be very low in experienced conditions. Herein magnetite is likely to be present. Authors mention that H₂ increasing preclude magnetite precipitation... this explain magnetite precipitation to stop but not the absence of magnetite before.

In order for magnetite to form, some of the ferrous iron dissolved from the olivine must be oxidized to Fe³⁺ by the reaction:

When olivine first dissolves, a tiny, unobservable amount of magnetite likely forms but as soon as the H₂ pressure in the fluid inclusion rises to the equilibrium value the reaction stops. The Fe²⁺ released from subsequent olivine dissolution is incorporated into amakinite [(Fe²⁺, Mg)(OH)₂]-brucite [Mg(OH)₂] solid solutions and iron-bearing serpentine [(Mg, Fe)₃Si₂O₅(OH)₄]. The only

way that more magnetite can form is for some of the H₂ to be lost from the inclusion by diffusion into the surrounding olivine crystal, allowing the reaction given above to continue. The reviewer's comment and question would apply if the phases dissolving and precipitating were pure Mg phases. As such, we have added the following sentence:

Line 108-113. "Small undetectable crystals of magnetite produced via reaction (2) should begin to precipitate inside the micro-reactors after reaction (1) starts and the solubility of magnetite in the solutions is exceeded. However, we interpret the lack of observable magnetite in the inclusions early in the experiment to reflect the fact that the crystals are not optically resolvable (~0.5 μm) by the methods used here."

In the Methods section

- The methods are well described and I estimate that it has sufficient detail, so that experiments and characterization can be reproduced.
- The discussion about uncertainties of the method to determine H₂O activity is not sufficiently detailed. e.g. what is the repercussion of an error of 0.1°C for FPD measurements (looking at supplementary table, comparing FIA1G and FIA1H SFI a difference of 0.1°C in TM change H₂O loss by a factor 1)

The uncertainty of 0.1 °C in the measured ice-melting temperature would have a different impact in the amount of H₂O consumed depending on the initial salinity. This is because the amount of H₂O that must be removed from solution to produce a 0.1 degree Celsius lowering of the FPD is also a function of the starting salinity (see Figure A below). Thus, if the initial salinity is 1 wt.% NaCl (FPD=-0.6°C), ~15% of the H₂O in the initial fluid must be consumed by hydration reactions to drive the FPD lower by 0.1 degree Celsius to -0.7°C (corresponding to a salinity of ~1.2 wt.% NaCl). However, the same 0.1 degree Celsius lowering of the FPD for a starting salinity of 10 wt.% NaCl requires consumption of only ~1.4 % of the initial H₂O in the fluid, as the FPD is lowered from -6.6°C (10 wt.% NaCl) to -6.7°C (10.1 wt.% NaCl). This is why the uncertainty of a 0.1 °C change in the FPD in experiments with 1 wt.% NaCl is higher than in the rest of the experiments. In other words the uncertainty of 0.1 °C in the FPD decreases as salinity increases.

To clarify the issue about uncertainty related to the precision of the ice melting temperature measurement, we added to the Methods section the following paragraph to explain this effect:

Line 461-482. "In our experimental method, we quantify the rates of serpentinization based on the amount of water removed from the aqueous solution and incorporated into hydrous phases. We use the freezing point depression (FPD) of the aqueous solution as a proxy for the amount of water removed from solution, and the precision of our FPD measurement is ±0.1°C. However, some amount of H₂O must be removed from the solution and incorporated into hydrous phases before the salinity is increased by an amount sufficient to produce a 0.1 degree Celsius lowering of the FPD. The actual amount of H₂O that must be removed from solution to produce a 0.1 degree Celsius lowering of the FPD is also a function of the starting salinity. Thus, for example, if the initial salinity is 1 wt.% NaCl (FPD=-0.6°C), ~15 percent of the H₂O in the initial fluid must be consumed by hydration reactions to drive the FPD lower by 0.1 degree Celsius to -0.7°C (corresponding to a salinity of ~1.2 wt.% NaCl). However, the same 0.1 degree Celsius lowering

of the FPD for a starting salinity of 3.5 wt.% NaCl requires the consumption of 4.7 percent of the H₂O in the initial fluid, 6 wt.% NaCl initial salinity requires a consumption of 2.6 percent of H₂O, and 10 wt.% NaCl initial salinity requires consumption of only 1.4% of the initial H₂O, as the FPD is lowered from -6.6°C (10 wt.% NaCl) to -6.7°C (10.1 wt.% NaCl). Thus, until a sufficient amount of reaction has occurred to consume enough H₂O to lower the FPD by ≥ 0.1 degree Celsius, in the absence of observed reaction products in the inclusions we have no way of knowing *a priori* that the reaction has started and, depending on the starting salinity, the amount of H₂O consumed (and, therefore, the amount of reaction that must occur) could require from ~1 to 15 percent of the initial H₂O to be removed from solution.”

Figure A. Relationship between the percent of H₂O that must be removed from the solution by the serpentinization reaction and the change in the freezing point depression for four different initial salinities.

- From what I understand, all SFI are assumed to be 10µm in diameter (impossible to measure single SFI volume), this is important to mention it more clearly as SFI initial volume might affect the rate at which salinity increases in fluid - even if it doesn't change measured salinity and inferred H₂O activity -. For a single sample even if initial salinity is the same in all SFI and reaction operate in the same way, olivine dissolution (related to surface, i.e. to V of SFI) remains a limiting factor (even when the steady state between dissolution and precipitation is reached), in turn H₂O consumption and salinity at a given time will be different due to different SFI initial volume and different "bulk SFI" reaction rate.

As the reviewer notes correctly, accurately determining the volume of a fluid inclusion is difficult and even though there are several ways to approximate the real volumes (Micro CT scans for example), our simplified approach is reasonable considering the size and shape distribution of the FI synthesized for our experiments; most of the FI are rounded to sub rounded with a range of sizes from mostly ~40 to 1 μm , and only few of the fluid inclusions followed here were irregular in shape. For practicality only FI > 5 μm were used in our study. We can assess whether our simplified visual approximation is reasonable by applying an averaging method (Tester et al., 1994) used in the past to determine the effective diameter of grain size when the sizes are distributed from two meshes. In this case we can think of the fluid inclusions as grains and the size distribution in the SFI goes from ~1 to 40 μm . Then,

$$D_e = \frac{D_{max} - D_{min}}{\ln\left(\frac{D_{max}}{D_{min}}\right)}$$

where D_e is the effective diameter (m), D_{max} is the maximum fluid inclusion size (40 μm) and D_{min} is the minimum inclusion size (1 μm). Then, the effective diameter of the fluid inclusions is ~10.6 μm , very close to our visual estimate.

In Figure 4A the rates are shown in log J mol/m²s for every activity of water (salinity). The dispersion of the rates span from ± 0.1 to 0.5 log units (mol/m²s). We infer that the data dispersion shown there and in the rates from Extended Table 4 (log r in mol/s) are most likely related to the variations in shapes and sizes (surface areas) of the FI.

For example, the volume of a 10 μm spherical inclusion is 523.6 μm^3 , and a sphere of this volume has a surface area of 314 μm^2 . Using this same volume (523.6 μm^3) and assuming a tabular (flat) FI with a thickness of 1 μm , corresponds to an upper and lower surface that is 23 x 23 μm . The surface area of the FI would be 23 x 23 x 2 (top and bottom surfaces) plus 23 x 1 x 4 (each of the four sides of the FI), for a total surface area of 1150 μm^2 .

If these two surface areas represent the two possible extremes of surface area for a given fluid inclusion volume and apply these to the average rate of 3.66×10^{-18} (mol/s) for all the experiments with the salinity 3.5 wt.% (Extended Table 4) we obtain:

Shape	Surface Area (m ²)	Rate (mol/s)	J (mol/m ² s)	Log J
Sphere	3.14E-10	3.67E-18	1.17E-08	-7.93
Parallelepiped	1.15E-09	3.67E-18	3.19E-09	-8.5

The difference in the log J for both shapes is ~0.5 log unit, well within the dispersions of the rates of ± 0.5 log units (log r in mol/s) shown in Extended data Table 4 and seen in Figure 4A. A similar dispersion of the data can be seen if the same calculation is made with the different sizes of the inclusion. For example:

Inclusion size	Surface Area (m ²)	Rate (mol/s)	J (mol/m ² s)	Log J
5 μm	7.85E-11	3.67E-18	4.67E-08	-7.33
10 μm	3.14E-10	3.67E-18	1.17E-08	-7.93

20 μm	1.26E-09	3.67E-18	2.92E-09	-8.5
------------------	----------	----------	----------	------

We agree that inclusion size and shape may affect the calculated rates that we are reporting, however they are within the observed dispersion of the measured rates (Extended data Table 4 and Figure 4A).

- Supplementary data are for me complete, but a detail of calculations for "H₂O consumed" in supplementary excel sheet will be appreciated.

We included as Supplementary Table 1 a spreadsheet with the detailed mass balance calculations used to determine "moles of H₂O consumed", as well as the fitting parameters used to calculate the moles of H₂O consumed by the changes in salinity of the FI.

-Figures and table are well chosen and well represent observations. Figure 1 schematic representation (1B) can have a better design with legend on one side and enlarged scheme. The legend can refer to Figures 2 and 3 as minerals fig. 1A are hard to see.

Fig. 1A has been modified to increase the size of the fluid inclusion images.

The summary of previous experimental studies is appropriate and well illustrate the importance of fluid salinity (as a proxy of water activity) on serpentinization kinetic at 280°C. The conclusion could mention the need to tests other temperatures, other water activities or the effect of other diluents (e.g. CO₂, SO₄...).

We agree with the suggestion and a sentence has been added to indicate this in line 181-183.

I recommend to publish this manuscript.

Dr. Romain Lafay

We thank Dr. Lafay for his overall positive comments. The comments and suggestion have greatly helped us to improve our manuscript.

Reviewer #2 (Remarks to the Author):

General comments

The study deals with the use of a novel and promising method to monitor fluid-mineral interactions in-situ. The method is based on the use of aqueous-fluid inclusions as micro-reactors. The authors have had the great idea to use the freezing point depression (FPD) of the fluid inclusion to derive its water content from salinity. Basically, if the host mineral is able to react with the water from the inclusion, as it is the case here for olivine forming hydrous phases (i.e., serpentine and brucite), reaction progress/kinetics can be inferred from time-resolved FPD measurements along with some extend of modeling. In comparison with conventional batch

experiments performed with sieved mineral grains, the present method avoids, in particular, artifacts associated with the presence of tiny little grains sticking onto the surface of the larger ones and which, consequently, escape sieving. These unexpected small grains often control reaction rates at low reaction extents.

This ingenious methodology has been applied to olivine serpentinization kinetics under increasing salinity (NaCl) from 1 to 10 wt.% at 280°C and ca. 500 bars. The authors observed a clear decrease in the serpentinization kinetics towards higher salinity / lower water activity. They therefore entitled their paper: "Effect of water activity on rates of serpentinization of olivine";. They conclude that serpentinization of oceanic lithosphere requires open-system behavior, i.e. with a reacting fluid which is renewed, otherwise serpentinization will cease due to increasing salinity and decreasing serpentinization rates.

I/ whereas I am very enthusiastic about the experimental method and its novelty, I find this main conclusion of the paper somehow trivial. Indeed, there are several pieces of evidence that serpentinization of the oceanic lithosphere is an open-system chemical process which is often described as an "alteration" process. Sr isotopic ratio ($^{87}\text{Sr}/^{86}\text{Sr}$), $\delta^{11}\text{B}$ or Li isotopes as well as changes in major element composition in serpentinites (e.g., Boschi et al. 2008, GCA 72, 1801-1823; Vils et al., EPSL 386, 414-425; Decitre et al., 2002, G3 3; Malvoisin, 2015, EPSL 430, 75-85) all support significant chemical transfer between "circulating" seawater and serpentinizing abyssal peridotites. The challenge today seems rather to quantify the magnitude of chemical transfers, e.g., through the derivation of water to rock ratio (W/R). Actually, one could argue that even in the hypothetical case where water would not be renewed in the serpentinization process, irrespective of the effect salinity on reaction kinetics, serpentinization would cease after the consumption of all the water. Therefore I would suggest that the authors use their remarkable dataset to put constraints on minimum W/R values for serpentinization to proceed at a realistic timescale (e.g., Früh-Green et al., 2003, Science 301, 495-498).

The reviewer states that the main conclusion of the paper is that serpentinization of oceanic lithosphere requires open-system behavior. While our results do, in fact, require open-system behavior in order for the serpentinization reaction to not shut down, we do not suggest that we have "discovered" this fact. Rather, we cite the paper by Rouméjon & Cannat (2014), which is one of many that have reported that open system behavior is required. As such, we have modified the text in the conclusions to state that our experimental results are consistent with previous models that suggest open system behavior (adding the suggested references), and our results also provide a geochemical explanation for why open system behavior is required. If the system is not recharged by lower-salinity fluids (i.e., seawater) the hydration reaction will progress until the salinity of the fluid reaches a point at which the high salinity and concomitant low water activity hinders further serpentinization. Furthermore, our results have major implications in the systems where fluid flow is restricted to grain boundary and nanoscale porosity (Plümper et al., 2012; Schwarzenbach et al., 2016; Tutolo et al., 2016).

To clarify these points we modified the text as follows:

Lines 175-183. "These results provide a geochemical basis to support the previously recognized notion that pervasive serpentinization of oceanic lithosphere (which is dominated by the mineral olivine) requires open-system behavior, as evidenced by the seawater-dominated isotope signature commonly observed in serpentinized peridotites (e.g., Boschi et al. 2008; Vils et al.,

2009; Decitre et al., 2002; Malvoisin, 2015). In particular, this requires continual influx of a lower salinity aqueous fluid (seawater) to dilute the serpentinization fluid and allow serpentinization of olivine to continue (Rouméjon & Cannat, 2014). Moreover, salinity may be a major rate-limiting factor where fluid influx is restricted due to the absence of open fractures and where fluid migration proceeds along grain boundaries or within nano-scale porosity (Plümper et al., 2012; Schwarzenbach et al., 2016; Tutolo et al., 2016).”

The reviewer further suggests that we relate our results to water-to-rock ratio (W/R). We do not think that using W/R ratios is reasonable owing to the ambiguity of the W/R concept. In fact, in describing W/R as applied to sub-seafloor interactions between seawater and basalt, Honnerez (in Rona et al., Hydrothermal Processes at Seafloor Spreading Centers) noted that *“The very concept of water:rock ratio is, to use Mottl’s own words, “necessarily a somewhat ambiguous” concept. Its definition varies with authors and terms such as “integrated” versus “instantaneous” ratios, “effective ratio”, or “static” ratio have been used with different significances.”* So, rather than relating our results to W/R, we believe that the new rate data and the additional data that could be derived for other temperatures and mineral systems would provide the rate laws that could be used for a quantitative (e.g. TOUGHREACT) model that would simultaneously consider reaction rate, permeability evolution, heat balance, water flow, etc. W/R values, if we would find an appropriate way to model them, are too simplified to correctly account for all of these processes that are occurring simultaneously during serpentinization associated with fluid flow.

2/ the authors nicely decomposed olivine serpentinization process into successive dissolution steps, the first of which being olivine dissolution (Reaction 3). However, recently, Olsen et al. (2015, JGR Planets 120, 388-400) showed how increasing ionic strength, IS (or lowering water activity) slows down olivine dissolution kinetics. The findings by Olsen et al. (2015) allow therefore to anticipate the effect of increasing IS on serpentinization kinetics as described here since olivine dissolution is expected to be the serpentinization rate-limiting step (e.g., Malvoisin et al., 2012). The results obtained here must be discussed in light of those published data.

We must first emphasize that one of the co-authors of the present paper, Don Rimstidt, was the MS advisor to Amanda Olsen at Virginia Tech and one of the co-authors in Olsen et al. (2015). We are thus familiar with the study of Olsen et al. We are encouraged to find that our results are consistent with the low temperature Olsen et al. study. However, Olsen et al. (2015) measured the rates of olivine dissolution (not serpentinization) at low pH, 25°C, 1 bar and fluid compositions (MgSO_4 , KNO_3 , $\text{Mg}(\text{NO}_3)_2$) and concentrations which are unrealistic for serpentinization reactions on Earth. The extreme extrapolation needed to project those rates to the high pH, high temperatures, and high pressures of serpentinization would introduce enormous errors. This extrapolation problem is avoided by our method of measuring the rates at the most common serpentinization conditions. Furthermore, the effect of activity of water on the dissolution rates of olivine reported in Olsen et al. study (activity coefficient of $n = 3.26$, Eq. 8 in Olsen et al. study) is dramatically different than the effect of activity of water observed in the serpentinization reactions of our study (activity coefficient of $n = 120.28$, considering a linear fit in to the data as Olsen et al. study or $n=103$ (Eq. 8 in our study) after the suggested modification of using a non linear regression). According to our data, the decrease in the rates of reaction as a function of salinity (activity of water) is two orders of magnitude greater compared with the

experiments at laboratory conditions of Olsen et al. study.

Nevertheless, we consider that it is appropriate to cite this and other studies that discuss the effect of salinity on dissolution kinetics (Schott et al., 2009; Casey & Westrich, 1992; Yardley and Bodnar, 2014; Morrow et al., 2014) and specially the Olsen et al. study because they did conclude that the dissolution of forsterite is a function of water activity, which in turn is controlled by the fluid salinity. As such, we have modified the text to cite that paper and discussed the Olsen et al. study, with the following sentence:

Lines 155-160. “The results show that serpentinization rates decrease rapidly with increasing salinity (or decreasing activity of water), and agree with previous studies that have reported that dissolution rates of forsterite at laboratory conditions (25 °C and 1 bar) decrease as $a_{\text{H}_2\text{O}}$ decreases³⁰”

And:

Line 166-174. “These results document the dramatic effect of salinity, as a proxy for water activity $a_{\text{H}_2\text{O}}$, on the rate of serpentinization of olivine and are consistent with previous studies that have suggested that salinity can have a significant effect on dissolution kinetics (Schott et al., 2009; Casey & Westrich, 1992; Yardley and Bodnar, 2014) including the dissolution of forsterite at far from equilibrium conditions (Morrow et al., 2014; Olsen et al., 2015). The effect of water activity $a_{\text{H}_2\text{O}}$ on the reaction rate is consistent with the generalized serpentinization reaction given by equation (1), in which the driving force for production of serpentine and brucite is inversely proportional to the cube of the activity of H_2O , and is also consistent with the dissolution of olivine (Equation 3) being the rate controlling step of the reaction (Malvoisin et al., 2012).”

More specific comments

1/ Reaction progress and pressure change

The authors based their kinetic approach on reaction rates $(\Delta n_w)/\Delta t$ what is obviously find. However, it would have been useful to present the kinetics data in terms of reaction progress as well. Basically, how much of the initial water content of the inclusion has been consumed over the reaction duration? Based on the initial inclusion volume of 5.24 10^{-10} cc (Line 362) with a density of 22.4 cc/mol (280°C/500 bar, supcrt92), there is about 2.3 10^{-11} mole H_2O per inclusion (NaCl content is not considered here). A water consumption of 1.2 10^{-11} mole as shown on extended data Fig.1A corresponds therefore to about 50% reaction progress, meaning that about half of the initial water has been consumed.

The suggested modifications have been included by adding extent of reaction (ξ). An extra column was added to Supplementary Table 1 and 2, adding the percent of H_2O loss and the extent of the reaction (ξ). Moreover, Extended Table 4 was also modified and a column containing the extent of the reaction (ξ) for each individual fluid inclusion was added. Furthermore, we included in Supplementary Table 1 a summary of how we calculated the extent of the reaction by fitting a function that correlates the salinity to the percent of H_2O consumed.

Another parameter of primary interest is the volume change of the inclusion as a function of reaction progress. Considering the molar volume @ 500 bar/280°C of the phases involved in Reaction (1), i.e., forsterite (43.8 cc/mol), brucite (24.6 cc/mol), chrysotile (108.5 cc/mole) and water (22.4 cc/mol), the volume change of reaction (1) is $-(21.7/3)$ cc per mole of H₂O consumed whereas the solid volume change is $+(45.5/3)$ cc/mole consumed H₂O (see Hanley, 1992, *Geology* 20, 705-708). For 50% reaction, the inclusion volume will decrease by $(45.5/3) \cdot 0.5 = 7.58$ cc, i.e. 35%. Accordingly, if 50% of the starting water has been consumed and if the inclusion volume has only decreased by 35%, this means that internal pressure has also decreased. Recalling that H₂O is liquid and thus very little compressible, we can anticipate a significant pressure drop, possibly down to the liq-vap curve! This very rough calculation is certainly not accurate but still the relationship between reaction progress, volume change and internal pressure must at least be discussed. The non-isobaric conditions in the micro-reactor may appear as a limitation of the technique.

Although these comments are interesting, and correct, they are not relevant to our method for determining the rates. In this study, rates were determined using the initial rate method, which fits the salinity (amount of water in the inclusion) versus time data to a function and then determines the slope of that function (= rate) at time = 0. At zero time little or no brucite + serpentine has formed and there is no significant water consumed. The volume and pressure changes that occur at a greater extent of reaction are not present and do not affect the result. Moreover, we should emphasize that measuring the freezing point depression depends on the ability to observe the changes inside the inclusion, though as reaction progresses the visibility in the inclusion is increasingly hindered (Lines 77-80). The initial rate method uses numerous short-term experiments that follow the reaction in the beginning stages. We agree with the reviewer that the non-isobaric conditions in the micro-reactors are a potential limiting factor. These limitations however, also overlap with the optical limitations of the technique, which in most cases made it impossible to follow the reaction to high extents of reaction. On average, we were only able to follow reaction progress up to about 33% ($\xi=0.33$) in all the experiments.

To further clarify these points we added the following text to the methods section:

Lines 520-539. "Reaction progress was followed in several dozen SFI by monitoring the change in the freezing point depression (FPD) through time. In order to follow the reaction progress it was necessary to observe the changes inside the SFI. As reaction progressed the amount of reaction products increased rendering the inclusion opaque and challenging our ability to follow the reaction progress at high extents of reaction. Moreover, since the H₂O inside the SFI is consumed by reaction (1), the internal pressure and volume, as well as the concentrations and composition inside the inclusion will be modified from the initial conditions. To avoid the problem of visibility and the effects of changing fluid compositions and pressure inside the SFI micro-reactors, our observations were incorporated into the Initial Rate Method (Burkin, 2001; Rimstidt, 2013). Batch experiments usually require data collection that include large extents of reaction to circumvent issues related to modifications of the initial conditions (i.e., composition, pH, pressure, etc.) (Rimstidt and Newcomb, 1993). Alternatively, the initial rate method uses numerous short-term experiments with a small number of concentration measurements at equally spaced times. Then, the concentration versus time data of each individual experiment are fitted to

a function to determine the slope of that function (= rate) at time = 0. In our experiments, at zero time little or no brucite + serpentine has formed and there is no significant water consumed. The volume and pressure changes that occur at a greater extent of reaction are not present and do not affect the result. As such, in our experiment, the number of moles of H₂O consumed by the serpentinization reaction with time was found from the change in n_w with time:”

2/ Effect of salinity

Another issue that could have been addressed with the present database is whether olivine dissolution kinetics differs in seawater and pure water. This is important since olivine serpentinization kinetics has been investigated experimentally in the past using either pure water or synthetic seawater. Here, the authors used a linear log-log regression to their H₂O flux - water activity data (Fig. 4A). Extrapolation of the regression to $a_{H_2O} = 1$ leads to a LogJ of -7.76 which, at 280°C, would be consistent with Wegner and Ernst’s dataset rather than with that of Malvoisin et al. (2012) as claimed Line 127 p. 6. A way to reconcile the most recent kinetics dataset with the present one would be to consider a non-linear regression, with $\log J = -8.5/-9.0$ at $a_{H_2O} = 1$. Then, differences between serpentinization kinetics in seawater (3.5 wt.%) and in pure water would cancel.

Through the course of this project several attempts to synthesis FI using pure water where conducted to use as a comparison. The resulting fluid inclusions formed after the experiments were too small (< 1 μm) so at the conditions of synthesis (650°C and 5.5 Kbar) no useful SFI were formed. The reviewer further suggests that one approach to reconcile different experimental data sets would be to use a non-linear regression instead of a linear regression for figure 4A. We accept this suggestion and we used a non-linear extrapolation to our rate equation instead of a linear regression. However, in doing so we discovered a small error in our previous calculations to convert from rates (mol/s) to fluxes (mol/m²s). The surface area was miscalculated by using the diameter of a fluid inclusion of 10 μm (1256.64 μm^2), instead of the radius of 5 μm (314 μm^2) as such our calculations where shifted by a factor of ~0.6 log unit (mol/m²s). This difference is not considered to be significant and does not change the overall conclusions of this study. We have corrected the mistake and changed the resulting rates in Extended Table 4.

The text was modified to accommodate these changes in the following manner:

Lines 158-165. A rate equation was generated by fitting the log of the reaction rates as a function of $\log a_{H_2O}$ with a second order polynomial:

$$\log J = -7.24(\pm 0.03) + 103(\pm 4)\log a_{H_2O} - 3481(\pm 4) ((\log a_{H_2O}))^2 \quad (8)$$

where -7.24 is the log of the rate constant at 280°C, and the coefficient 103 represents the apparent reaction order of H₂O. The serpentinization rates obtained here agree with the rates of reaction from Wegner and Ernst (1983) after extrapolating the results to activity of 1 (pure water) with a Log J of -7.51(Fig. 4B).

We thank the reviewer #2 for the positive comments, as well as the suggestions that have greatly helped us to improve our manuscript.

Reviewer #3 (Remarks to the Author):

Review of manuscript NCOMMS-16-22733-T by Lamadrid et al.

This manuscript provides rates of olivine serpentinization as a function of fluid salinity, a parameter that has not been investigated before, most of the previous studies using either pure water or seawater salinities. The experimental protocol used here is original, as it associates previously developed techniques (synthetic fluid inclusion formation, salinity measurements classically used for fluid inclusion studies, optical/Raman monitoring used e.g. in diamond anvil cells, and final FIB-TEM/SEM and Raman characterization of reaction products). TOF-SIMS is also used and provides valuable chemical analyses that locate the different minerals and the fluid within the fluid inclusion, as it may have been done with Raman mapping.

The paper is concise and well written. Results are undoubtedly of good quality, consistent and provide new data on serpentinization rate that complement previously published ones. But despite the strong interest of the community for serpentinization, the conclusions on the role of water activity on serpentinization remains limited as water is an essential ingredient to this hydration reaction and its activity is of course determinant. Implications on the required open-system behavior and the fresh fluid renewal at olivine interface, notably by cracking/fracturation, are not very new neither. This has been proposed by many other authors from various types of studies (Jamveit et al., Andreani et al. 2007, Plummer et al., 2012; Godard et al, 2013; Klein et al. 2015, ...) - it is already very well established. The technical development, while elegant, is inspired from previous methods and does not appear sufficient to justify the publication of this paper in a journal as Nature Comm.

We respectfully disagree with the conclusion of this reviewer. While many others have noted that open system behavior is required to sustain the serpentinization reaction, to our knowledge no previous studies have quantified the relationship between water activity and reaction rates at PT conditions appropriate for serpentinization in the sub-seafloor. In addition, though the importance of open-system behavior has been suggested previously, previous studies have looked at it from a different perspective. In particular, we consider that salinity is an essential control where fluid influx is limited (e.g., where open fractures are absent) and where fluids migrate along grain boundaries or within nano-scale porosity. Moreover, it is unclear why the reviewer states that the method is inspired from previous methods. There are currently no experimental methods that 1) permit monitoring the reaction in real time and *in situ*, and 2) that allow us to measure accurately the amounts of reactants produced using both optical and various spectroscopic techniques. From one experiment, using 1 or 2 crystals, we can obtain several wafers (6 to 8) that contain hundreds of SFIs that can be used as micro-reactors. This is a novel, simple and elegant approach that integrates decades of combined proven research from the fields of experimental petrology, fluid inclusions and geochemical kinetics (chemical reactors). As Dr. Lafay and Reviewer #2 have stated, they believe that this is a promising, innovative and ingenious technique that could be applied to other mineral and fluid systems and at different PVT conditions. As such, it appears to us and to two of the reviewers that this work appeals to a multi-disciplinary audience that goes beyond the geosciences, biology and chemical engineering.

I also have some other questions/comments on the manuscript that deserve improvements:

- How is the volume of fluid inclusion measured? Not clear.

The volume of the fluid inclusions was not measured directly. As answered in the comments of reviewer #1 we apply a simplified approach and use the volume of a spherical inclusion of 10 microns as our starting volume.

To clarify this we added in line 394:

Line 490-491. “where V_{SFI} is the volume of an ideal spherical fluid inclusion ($= 5.24 \cdot 10^{-10} \text{ cm}^3$) and ρ_{SFI} is the density of the solution in the SFI (in g/cm^3).”

- The reactive surface area is calculated for a spherical inclusion which is obviously not the case according to images. What is the error on rate calculations?

Same comments as for Reviewer #1; the overall uncertainty in the rate calculations using different sizes and shapes of the fluid inclusions are of ~ 0.5 log units $\text{mol/m}^2\text{s}$, this uncertainty is within the observed dispersion of the measured rates (Extended data Table 4 and Figure 4A).

- Line 66: Why only 50% of the SFI reacted after 15 days for a salinity of 1 wt% if they are all similar? What happened in the others? Reaction is delayed?

The overall reaction involves the simultaneous dissolution of olivine and the precipitation of serpentine + brucite. The observed lag time for the onset of the salinity change depends upon the incubation time needed for nucleation of the serpentine + brucite reaction products. Differences in this incubation time appear to be caused by different abundances of heterogeneous nucleation sites (defects) on the olivine surfaces.

- Raman identification of hydrogen indicate a gaseous phase, right? Is it only due to cooling during measurement? Or do you also see it in your isothermal experiments?

Hydrogen was observed in the micro-reactors in the liquid phase during the isothermal experiments, and in the gas phase after cooling to room T. Raman mapping was conducted at room T and the hydrogen was observed in the vapor phase, as shown in Figure 2.

- Line 83-87: Why is it necessary to remove hydrogen? Is the equilibrium reached? Does this sentence result from any calculation, e.g. based on magnetite grain size over hydrogen solubility and FI volume, or other?

Once the hydrogen pressure (fugacity) reaches equilibrium as shown in reaction (2), additional magnetite will not form as the forward and back reactions will be equal. The amount of magnetite that forms up to the point at which the hydrogen pressure reaches the equilibrium value would be negligible (and would not be recognized in the inclusions).

So, in order for the reaction to continue to make more magnetite, hydrogen must be removed to force the reaction to the right.

- How is hydrogen retained in some natural fluid inclusions if it already diffuses at lab timescale?

The occurrence of H_2 in natural inclusions is rare but it has been reported. It is found in FI associated with very high-grade uranium deposits, such as the Oklo (Gabon) deposit (Dubessy et al. 1988). In this case, it is thought that the hydrogen is generated as a result of radiolysis of water from the intense nuclear radiation. Hydrogen is also rarely reported along with methane in some FI in alkalic intrusive rocks. In all cases, it is thought that reactions *in situ* are generating hydrogen in the FI.

One case of hydrogen reported as trapped and not produced *in situ* is from Kelley et al. (1996) who report H_2 in H_2O -rich fluid inclusions containing CH_4 and CO_2 in plagioclase from dredged samples from the Southwest Indian Ridge. The reason hydrogen is still contained in those inclusions is likely because of the cold temperatures at which the inclusions were trapped and then recovered (bottom of the ocean). The experimental data available for diffusion of hydrogen in nominally anhydrous minerals is mostly conducted at magmatic temperatures (e.g. Demuchey & Mackwell, 2006), where diffusion occurs fairly fast. However, an extrapolation of the data to the conditions of our experiments for olivine (see Figure B below) shows that the diffusion is 12 orders of magnitude slower. If we extrapolate even further to the temperatures at which the samples from Kelley et al. were collected (dredged from the ocean floor) the diffusion rates are around 30 to 40 orders of magnitude slower.

It is important to note that we are not suggesting that this extrapolation is by any means correct but it can serve as an analogy to what the temperature conditions of trapping and exhumation could have been in order to still find H_2 inside the fluid inclusions reported by Kelley et al.

Another possibility that could explain the presence of H_2 in fluid inclusions in minerals that contain Fe in their composition (olivine and pyroxenes) could be by the oxidation of Fe^{2+} to Fe^{3+} as a reaction between the host and the trapped fluid.

Figure B. Model for the diffusivity of hydrogen through olivine from Demouchy & Mackwell (2006) as a function of temperature. A) The diffusivity models parallel to the crystal plane (001) in red and for plane (100,10) in blue as a function of temperature. The equation (solid line) was extrapolated (dashed line) to lower temperatures to estimate the rate at which hydrogen will diffuse out of the SFI at 280°C. For the T of our experiments the diffusivity of H₂ in olivine is between 12 to 14 orders of magnitude slower than at higher T.

- It would be interesting on figure 4 to report on the salinity of the previous experiments for a direct comparison.

In the figure caption of Figure 4 we added the salinities of each work. We also added the following text.

Lines 631-636. “Most of the previous experimental studies have used pure water (Martin & Fyfe, 1970; Wegner & Ernst, 1983; Ogasawara et al., 2013; Malvoisin et al., 2012). Malvoisin et al. (2012) conducted one experiment using a fluid with salinity analogous to seawater. Lafay et al. (2012) used a fluid with a concentration of 1 M of NaOH. McCollom et al. (2016) used a fluid with a concentration of 485 mmol NaCl/kg and 19.4 mmol NaHCO₃/kg to approximate seawater salinity and to provide a source of carbon.”

Reviewers' Comments:

Reviewer #2:

Remarks to the Author:

My general concern, which is actually shared by Reviewer#3, lies in the identification of the scientific novelty in this paper that would deserve publication in Nature Comm. The authors do agree that "chemical openness" with respect to the natural serpentinization process is not a new idea. They also explain that the effect of lowering water activity on aqueous reaction kinetics involving dissolution/crystallization processes has already been demonstrated. As I already mentioned in my previous review, the experimental approach that has been developed here is ingenious and innovative. Therefore, beyond its successful use to the study of the a_{H_2O} dependent serpentinization kinetics, the new perspectives that it opens with respect to the study of hydrothermal processes could rather form the heart of this paper.

Beside this general impression, I acknowledge the very good work performed by the authors at addressing my more technical comments. I feel that the ms has significantly gained in clarity.

Comments from Reviewer #2 (Remarks to the Author):

My general concern, which is actually shared by Reviewer#3, lies in the identification of the scientific novelty in this paper that would deserve publication in Nature Comm. The authors do agree that “chemical openness” with respect to the natural serpentinization process is not a new idea. They also explain that the effect of lowering water activity on aqueous reaction kinetics involving dissolution/crystallization processes has already been demonstrated. As I already mentioned in my previous review, the experimental approach that has been developed here is ingenious and innovative. Therefore, beyond its successful use to the study of the aH₂O dependent serpentinization kinetics, the new perspectives that it opens with respect to the study of hydrothermal processes could rather form the heart of this paper.

Beside this general impression, I acknowledge the very good work performed by the authors at addressing my more technical comments. I feel that the ms has significantly gained in clarity.

Response:

We concur that the concept that serpentinization requires open system behavior is not new. Our results support this earlier conclusion. We thank the reviewer for his/her kind words concerning the quality of the work.